# Sustained bacterial N₂O reduction at acidic pH

Guang He ®[1,2], Gao Chen ®[2,3], Yongchao Xie[2,7], Cynthia M. Swift ®[2,3], Diana Ramirez[4,5], Gyuhyon Cha[6], Konstantinos T. Konstantinidis ®[6], Mark Radosevich[1] & Frank E. Löffler ®[1,2,3,4,5] ✉

Nitrous oxide ($N_2O$) is a climate-active gas with emissions predicted to increase due to agricultural intensification. Microbial reduction of $N_2O$ to dinitrogen ($N_2$) is the major consumption process but microbial $N_2O$ reduction under acidic conditions is considered negligible, albeit strongly acidic soils harbor *nosZ* genes encoding $N_2O$ reductase. Here, we study a co-culture derived from acidic tropical forest soil that reduces $N_2O$ at pH 4.5. The co-culture exhibits bimodal growth with a *Serratia* sp. fermenting pyruvate followed by hydrogenotrophic $N_2O$ reduction by a *Desulfosporosinus* sp. Integrated omics and physiological characterization revealed interspecies nutritional interactions, with the pyruvate fermenting *Serratia* sp. supplying amino acids as essential growth factors to the $N_2O$-reducing *Desulfosporosinus* sp. Thus, we demonstrate growth-linked $N_2O$ reduction between pH 4.5 and 6, highlighting microbial $N_2O$ reduction potential in acidic soils.

pH is a key parameter controlling soil biogeochemistry, but soil acidification, a natural process accelerated by the reliance of synthetic nitrogen fertilizer, the growth of legumes, and acidic precipitation/deposition, plagues regions around the world[1]. Biological processes fix about 180 Tg N per year[2] and conventional agriculture introduces more than 100 Tg N of chemically fixed N each year[3]. N input accelerates soil N cycling resulting in increased formation of $N_2O$, a compound linked to ozone depletion and climate change[4,5], as well as to the inhibition of biogeochemical processes such as methanogenesis, mercury methylation, and reductive dechlorination[6–8]. The rise in global $N_2O$ emissions indicates an imbalance between $N_2O$ formation versus consumption, which has been attributed to the functionality of the resident microbiome[9] and environmental variables including the availability of electron donors for N oxide reduction[10–12], the concentrations of N oxyanions[13], oxygen content[14,15], copper availability[16,17], and pH[18]. The reduction of $N_2O$ to environmentally benign $N_2$ appears particularly susceptible to acidic pH, and acidic environments are generally considered $N_2O$ emitters[19–23]. A few studies reported $N_2O$ consumption in denitrifying soil (slurry) microcosms with pH values below 5[20,24,25]; however, soil heterogeneity and associated microscale patchiness of pH conditions, as well as pH increases during the incubation, make generalized conclusions untenable[26,27]. Attempts with denitrifying enrichment and axenic cultures derived from soil have thus far failed to demonstrate growth-linked $N_2O$ reduction and associated sustainability of such a process under acidic (pH < 6) conditions[27–29].

The only known sink for $N_2O$ are microorganisms expressing $N_2O$ reductase (NosZ), a periplasmic, copper-containing enzyme that catalyzes the conversion of $N_2O$ to environmentally benign dinitrogen ($N_2$). NosZ expression and proteomics studies with the model denitrifier *Paracoccus denitrificans* suggested that acidic pH interferes with NosZ maturation (e.g., copper incorporation into two dinuclear centers, $Cu_Z$ and $Cu_A$)[30,31], a phenomenon also observed in enrichment cultures harboring diverse $N_2O$-reducing bacteria[32]. Studies with

[1]Department of Biosystems Engineering and Soil Science, The University of Tennessee, Knoxville, Knoxville, TN 37996, USA. [2]Department of Civil and Environmental Engineering, The University of Tennessee, Knoxville, Knoxville, TN 37996, USA. [3]Center for Environmental Biotechnology, The University of Tennessee, Knoxville, Knoxville, TN 37996, USA. [4]Department of Microbiology, The University of Tennessee Knoxville, Knoxville, TN 37996, USA. [5]Biosciences Division, Oak Ridge National Laboratory, Oak Ridge, TN 37831, USA. [6]School of Civil and Environmental Engineering, Georgia Institute of Technology, Atlanta, GA 30332, USA. [7]Present address: Department of Chemistry and Biochemistry, University of California, Los Angeles, Los Angeles, CA 90095, USA. ✉e-mail: frank.loeffler@utk.edu

*Marinobacter hydrocarbonoclasticus* found active NosZ with a $Cu_Z$ center in the 4Cu2S form in cells grown at pH 7.5, but observed a catalytically inactive NosZ with the $Cu_Z$ center in the form 4Cu1S when the bacterium was grown at pH 6.5[33]. The inability to synthesize functional canonical NosZ serves as explanation for increased $N_2O$ emissions from acidic pH; however, this paradigm cannot explain $N_2O$ consumption in acidic soils[34,35].

A metagenome-based analysis of soil microbial communities in the Luquillo Experimental Forest (El Yunque National Forest, Puerto Rico) provided evidence that $N_2O$-reducing soil microorganisms are not limited to circumneutral pH soils and exist in strongly acidic (pH 4.5-5.0) tropical forest soils[36]. Anoxic microcosms established with acidic Luquillo Experimental Forest soil and maintained at pH 4.5 demonstrated sustained $N_2O$ reduction activity, and comparative metagenomic studies implicated strict anaerobic taxa harboring clade II *nosZ*, but lacking nitrite reductase genes (*nirS*, *nirK*), in $N_2O$ reduction[37]. While the effects of pH on facultative anaerobic, denitrifying species have been studied[30,32,38], efforts to explore strict anaerobic non-denitrifiers capable of $N_2O$ reduction are largely lacking.

In this work, we integrate cultivation and omics approaches to characterize a non-denitrifying two-species co-culture derived from acidic tropical soil. The co-culture comprises an acidophilic, anaerobic bacterium, *Desulfosporosinus nitrosoreducens*, that couples respiratory

$N_2O$ reduction with hydrogen oxidation at pH 4.5 – 6.0, but not at or above pH 7.

## Results

### A consortium consisting of two species reduces $N_2O$ at pH 4.5

Microcosms established with El Verde tropical soil amended with lactate consumed $N_2O$ at pH 4.5; however, $N_2O$-reducing activity was lost upon transfers to vessels with fresh medium containing lactate. The addition of acetate, formate (1 or 5 mM each), and $CO_2$ (208 μmol, 2.08 mM nominal), propionate (5 mM), or yeast extract (0.10 – 10 g L$^{-1}$) did not stimulate $N_2O$ reduction in pH 4.5 transfer cultures. Limited $N_2O$ consumption was observed in transfer cultures amended with 2.5 mM pyruvate, but complete removal of $N_2O$ required the addition of $H_2$ or formate. In transfer cultures with $H_2$ or formate, but lacking pyruvate, $N_2O$ was not consumed. Subsequent transfers in completely synthetic basal salt medium amended with both pyruvate and $H_2$ yielded a robust enrichment culture that consumed $N_2O$ at pH 4.5 (Fig. 1). Phenotypic characterization illustrated that pyruvate utilization was independent of $N_2O$, while $N_2O$ reduction only commenced following pyruvate consumption. The fermentation of pyruvate yielded acetate, $CO_2$, and formate as measurable products, with formate and external $H_2$ serving as electron donors for subsequent $N_2O$ reduction (Supplementary Fig. 1 and Note 1). The fermentation of pyruvate resulted in pH increases, with the magnitude of the medium

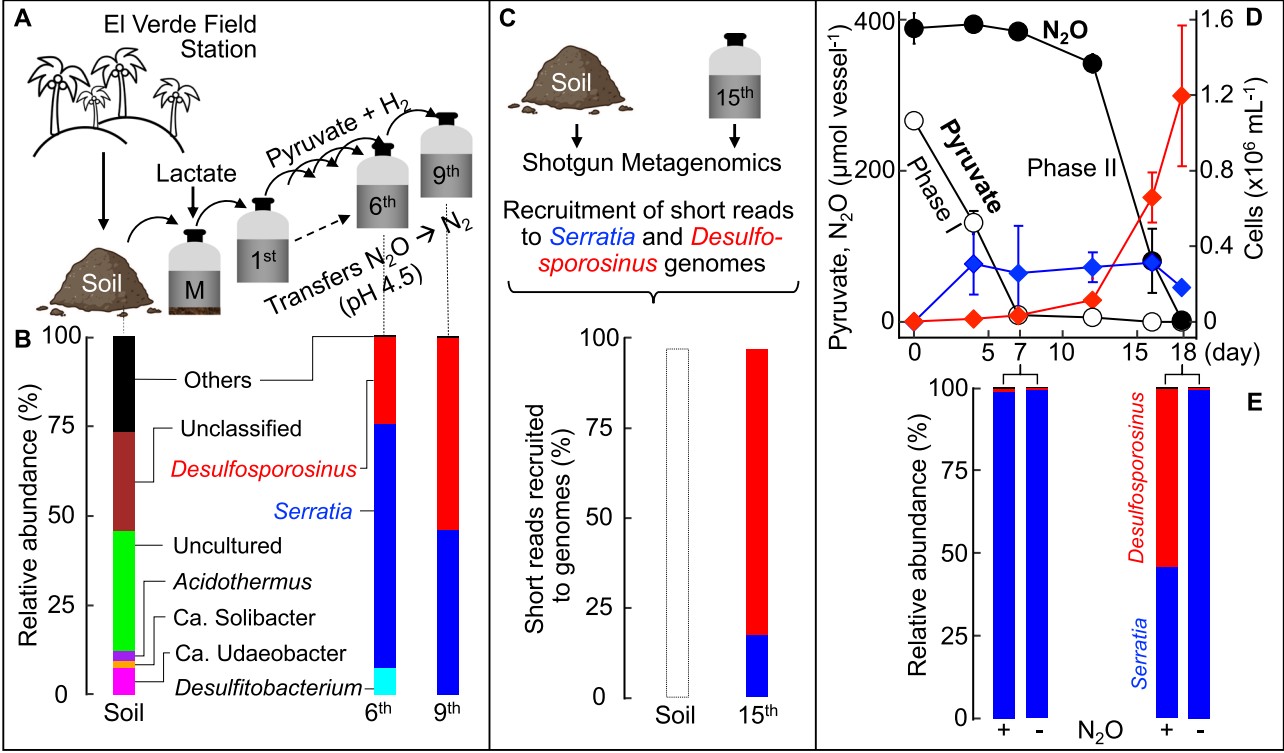

**Fig. 1 | Establishment of low pH $N_2O$-reducing microcosms and enrichment cultures yielding co-culture EV sourced from El Verde tropical soil. A** Schematic of the workflow leading from a soil sample to a solids-free enrichment, and to a co-culture. **B** Community structure based on 16S rRNA gene sequence analysis documents the enrichment process. Profiling of the soil microbial community was based on 16S rRNA genes extracted from shotgun metagenomic reads. Community profiling of 6th and 9th transfer cultures was based on 16S rRNA gene amplicon sequencing. Sequences with abundances <2% were grouped as "Others". **C** Percent of the metagenomic short read fragments obtained from El Verde soil and the 15th transfer culture that recruited to the genomes of *Serratia* sp. or *Desulfosporosinus* sp. A representative graph showing the high identity (> 95%) of reads mapping evenly across the *Desulfosporosinus* sp. genome is presented in Supplementary Fig. 11. The *Serratia* sp. or *Desulfosporosinus* sp. genomes were not detected in the

soil metagenome dataset, with less than 0.01% of the metagenomic reads mapping to the two genomes (white bar). **D** Pyruvate fermentation (Phase I, white circles) and $N_2O$ consumption (Phase II, black circles) in co-culture EV. *Serratia* (blue diamonds) and *Desulfosporosinus* (red diamonds) cell numbers were determined with specific, 16S rRNA gene-targeted qPCR. **E** Amplicon sequencing illustrates the population shifts in co-culture EV following pyruvate consumption (day 7) and following $N_2O$ consumption (day 18). Relative abundance of *Serratia* (blue bars) and *Desulfosporosinus* (red bars) in co-culture EV following Phase I (day 7) and Phase II (day 18) in vessels with pyruvate plus $N_2O$ (left) and pyruvate only (right). Representative cultures were sequenced. All other data represent the averages of triplicate incubations and error bars represent standard deviations (n = 3). Error bars are not shown if smaller than the symbol. Source data are provided as a Source Data file. The soil image was created with BioRender.com.

pH change proportional to the initial pyruvate concentration. The fermentation of 2.5 mM pyruvate increased the medium pH by $0.53 \pm 0.03$ pH units whereas a lower pH increase of $0.22 \pm 0.02$ pH units was observed with 0.5 mM pyruvate (Supplementary Fig. 2). $N_2O$ reduction was also observed in cultures that received 5 mM glucose. $N_2O$ reduction was oxygen sensitive and $N_2O$ was not consumed in medium without reductant (i.e., cysteine or dithiothreitol).

Microbial community profiling of El Verde soil and solids-free transfer cultures documented effective enrichment in defined pH 4.5 medium amended with pyruvate, $H_2$, and $N_2O$ (Fig. 1B and Supplementary Note 2). Following nine consecutive transfers, *Serratia* and *Desulfosporosinus* each contributed about half of the 16S rRNA amplicon sequences (49.7% and 50.2%, respectively), and less than 0.05% of the sequences represented *Planctomycetota, Lachnoclostridium, Caproiciproducens*.

Deep shotgun metagenome sequencing performed on a 15[th] transfer culture recovered two draft genomes representing the *Serratia* sp. and the *Desulfosporosinus* sp., accounting for more than 95% of the total short read fragments. All 16S rRNA genes associated with assembled contigs could be assigned to *Serratia* or *Desulfosporosinus* (Supplementary Fig. 3 and Note 2), indicating that the enrichment process yielded a consortium consisting of a *Serratia* sp. and a *Desulfosporosinus* sp., designated co-culture EV (El Verde). Efforts to recover the *Serratia* and *Desulfosporosinus* genomes from the original soil metagenome data sets via recruiting the soil metagenome fragments to the two genomes (Fig. 1C) were not successful, highlighting the effectiveness of the enrichment strategy. Redundancy-based analysis with Nonpareil[39] revealed that the average covered species richness in the metagenome data set obtained from the 15[th] transfer culture was 99.9%, much higher than what was achieved for the El Verde original soil inoculum (39.5%), suggesting the metagenome analysis of the original soil did not fully capture the resident microbial diversity.

The application of 16S rRNA gene-targeted qPCR assays to DNA extracted from 9[th] transfer $N_2O$-reducing cultures revealed a bimodal growth pattern. During pyruvate fermentation (Phase I), the *Serratia* cell numbers increased nearly 1,000-fold from $(2.3 \pm 0.8) \times 10^2$ to $(1.8 \pm 0.2) \times 10^5$ cells mL$^{-1}$, followed by a 40-fold increase from $(3.5 \pm 1.5) \times 10^4$ to $(1.2 \pm 0.4) \times 10^6$ cells mL$^{-1}$ of *Desulfosporosinus* cells during $N_2O$ reduction (Phase II) (Fig. 1D). In vessels without $N_2O$, *Desulfosporosinus* cell numbers did not increase, indicating that growth of this population depended on the presence of $N_2O$. Growth yields of $(3.1 \pm 0.11) \times 10^8$ cells mmol$^{-1}$ of $N_2O$ and $(7.0 \pm 0.72) \times 10^7$ cells mmol$^{-1}$ of pyruvate were determined for the *Desulfosporosinus* and the *Serratia* populations, respectively. The growth yield of *Desulfosporosinus* with $N_2O$ as electron acceptor is on par with growth yields reported for neutrophilic $N_2O$-reducing bacteria with clade II *nosZ* under comparable growth conditions[40,41]. 16S rRNA gene amplicon sequencing performed on representative samples collected at the end of Phase I (day 7) and Phase II (day 18) confirmed a bimodal growth pattern. Sequences representing *Serratia* increased during Phase I and *Desulfosporosinus* sequences increased during Phase II (Fig. 1E). Taken together, the physiological characterization, qPCR, genomic, and amplicon sequencing results indicate that co-culture EV performs low pH $N_2O$ reduction, with a *Serratia* sp. fermenting pyruvate and a *Desulfosporosinus* sp. reducing $N_2O$. Streaking aliquots of a 1:10-diluted 15[th] co-culture suspension sample onto Tryptic Soy Agar (TSA) solid medium under an air headspace yielded an axenic *Serratia* sp., designated strain MF, capable of pyruvate fermentation. Despite extensive efforts, the $N_2O$-reducing *Desulfosporosinus* sp. resisted isolation, presumably due to obligate interaction(s) with strain MF (see below and Supplementary Note 3).

## Identification of auxotrophies

To investigate the specific nutritional requirements of the *Desulfosporosinus* sp. in co-culture EV, untargeted metabolome analysis was conducted on supernatant collected from axenic *Serratia* sp. cultures growing with pyruvate and during $N_2O$ consumption (Phase II) following inoculation with co-culture EV (Fig. 2A). Peaks representing potential metabolites were searched against a custom library (Supplementary Dataset 1) and 33 features could be assigned to known structures, including seven amino acids (alanine, glutamate, methionine, valine, leucine, aspartate, and tyrosine). Cystine, the oxidized derivative of the amino acid cysteine, was also detected; however, cystine or cysteine were not found in cultures where dithiothreitol (DTT) replaced cysteine as the reductant, suggesting that *Serratia* did not excrete either compound into the culture supernatant. Time series metabolome analysis of culture supernatant demonstrated dynamic changes to the amino acid profile following inoculation with the *Serratia* sp. and the *Desulfosporosinus* sp. (as co-culture EV) (Fig. 2A, B). Alanine, valine, leucine, and aspartate increased during pyruvate fermentation (Phase I) and were not consumed by the *Serratia* sp. (Supplementary Fig. 4). Consumption of alanine, valine, leucine, and aspartate did occur following the inoculation of the *Desulfosporosinus* sp. (as co-culture EV) (Fig. 2A). These findings suggest that the $N_2O$-reducing *Desulfosporosinus* sp. is an amino acid auxotroph, and a series of growth experiments explored if amino acid supplementation (Supplementary Table 1) could substitute the requirement for pyruvate fermentation by the *Serratia* sp. for enabling $N_2O$ consumption by the *Desulfosporosinus* sp. The addition of individual amino acids ($n = 20$) was not sufficient to initiate $N_2O$ reduction in pH 4.5 medium, as was the combination of alanine, valine, leucine, aspartate, and tyrosine. Incomplete $N_2O$ consumption (<20% of initial dose) was observed in cultures supplemented with the 5-amino acid combination plus methionine. $N_2O$ reduction and growth of the *Desulfosporosinus* sp. occurred without delay in cultures supplied with a 15-amino acid mixture (Fig. 2C). Omission of single amino acids from the 15-amino acid mixture led to incomplete $N_2O$ reduction, similar to what was observed with the 6-amino acid combination. Efforts to isolate the *Desulfosporosinus* sp. in medium without pyruvate but amended with amino acids were unsuccessful because of concomitant growth of the *Serratia* sp., as verified with qPCR.

## pH range of acidophilic $N_2O$ reduction by the *Desulfosporosinus* sp

Growth assays with co-culture EV were performed to determine the pH range for $N_2O$ reduction. Co-culture EV reduced $N_2O$ at pH 4.5, 5.0 and 6.0, but not at pH 3.5, 7.0 and 8.0. pH 4.5 cultures exhibited about two times longer lag periods (i.e., 10 versus 5 days) prior to the onset of $N_2O$ consumption than cultures incubated at pH 5.0 or 6.0 (Supplementary Fig. 5A). In medium without amino acid supplementation, pyruvate fermentation was required for the initiation of $N_2O$ consumption (Fig. 1C), raising the question if pH impacts pyruvate fermentation by the *Serratia* sp., $N_2O$ reduction by the *Desulfosporosinus* sp., or both processes. Axenic *Serratia* sp. cultures fermented pyruvate over a pH range of 4.5 to 8.0, with the highest pyruvate consumption rates of $1.47 \pm 0.04$ mmol L$^{-1}$ day$^{-1}$ observed at pH 6.0 and 7.0, and the lowest rates measured at pH 4.5 ($0.43 \pm 0.05$ mmol L$^{-1}$ day$^{-1}$) (Supplementary Fig. 5B). The $N_2O$ consumption rates in co-culture EV between pH 4.5 to 6.0 were similar and ranged from $0.24 \pm 0.01$ to $0.26 \pm 0.01$ mmol L$^{-1}$ day$^{-1}$ (Supplementary Fig. 5C). These findings suggest that pyruvate fermentation by *Serratia* sp., not $N_2O$ reduction by *Desulfosporosinus* sp., explains the extended lag periods observed at pH 4.5 (Supplementary Fig. 5A). Consistently, shorter lag phase for both $N_2O$ reduction and *Desulfosporosinus* growth were observed in co-culture EV amended with the amino acid mixture (Fig. 2C).

## Phylogenomic analysis

Phylogenomic reconstruction based on concatenated alignment of 120 bacterial marker genes corroborated the affiliation of the $N_2O$-reducing bacterium with the genus *Desulfosporosinus* (Fig. 3). The

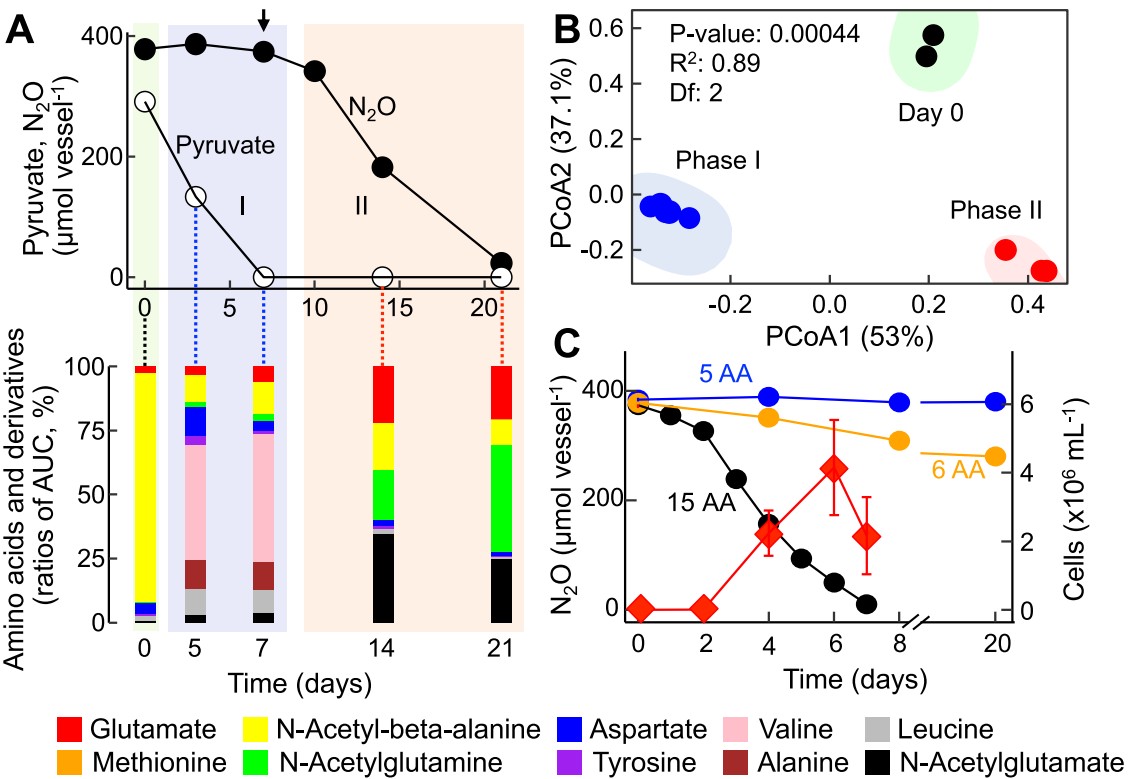

**Fig. 2 | Interspecies cross-feeding supports low pH N$_2$O reduction in co-culture EV. A** Pyruvate fermentation (Phase I, blue background) in vessels inoculated with axenic *Serratia* sp. strain MF and N$_2$O consumption (Phase II, red background) following inoculation (indicated by the arrow) with co-culture EV comprising strain MF and the N$_2$O-reducing *Desulfosporosinus* sp. The bottom part of (**A**) shows the amino acid profile in the supernatant immediately after inoculation with strain MF (green background), during Phase I, and during Phase II following inoculation with co-culture EV (day 7; 3% inoculum). Samples for untargeted metabolome analysis were collected immediately after inoculation with strain MF (green background), during Phase I (blue) and Phase II (red). The stacked bars show the ratio (%) of areas under the curve (AUC) of the respective amino acids and amino acid derivatives. Metabolites not assigned to structures representing amino acids or its derivatives are not shown. **B** Principal coordinate analysis (PCoA) of amino acid profiles. The enclosing ellipses were estimated using the Khachiyan algorithm with the ggforce package. Two-sided Permanova analysis was conducted with 99,999 permutations using the vegan Community Ecology package. Black, blue, and red circles represent samples collected at day 0, during Phase I, and during Phase II, respectively. **C** N$_2$O consumption in co-culture EV in medium amended with mixtures comprising 5 (blue), 6 (orange), or 15 (black) amino acids (Supplementary Table 1), H$_2$, and N$_2$O. The *Desulfosporosinus* sp. cell numbers (red diamonds) were determined with 16S rRNA gene-targeted qPCR and show growth in medium receiving the 15-amino acid mixture. Various other amino acid mixtures tested resulted in no or negligible N$_2$O consumption. All growth assays with amino acid mixture supplementation were performed in triplicates and repeated in independent experiments. The data shown in Fig. 2C represent the averages of triplicate incubations and error bars represent the standard deviations. Error bars are not shown when smaller than the symbol. Source data are provided as a Source Data file.

genus *Desulfosporosinus* comprises strictly anaerobic, sulfate-reducing bacteria, and *Desulfosporosinus acididurans* strain SJ4 and *Desulfosporosinus acidiphilus* strain M1 were characterized as acidophilic sulfate reducers. Genome analysis revealed shared features between the N$_2$O-reducing *Desulfosporosinus* sp. and characterized *Desulfosporosinus* spp. (Supplementary Note 4). The N$_2$O-reducing *Desulfosporosinsus* sp. in co-culture EV possesses the *aprAB* and *dsrAB* genes encoding adenylyl sulfate reductase and dissimilatory sulfate reductase, respectively, but lacks the *sat* gene encoding sulfate adenylyl-transferase/sulfurylase. To provide experimental evidence that the N$_2$O-reducing *Desulfosporosinus* sp. in co-culture EV lacks the ability to reduce sulfate, a hallmark feature of the genus *Desulfosporosinus*, comparative growth studies were performed. The N$_2$O-reducing *Desulfosporosinsus* sp. in co-culture EV did not grow with sulfate as sole electron acceptor, consistent with an incomplete dissimilatory sulfate reduction pathway (Supplementary Fig. 6A). *Desulfosporosinus acididurans* strain D[42], a close relative of the N$_2$O-reducing *Desulfosporosinus* sp. in co-culture EV, grew with sulfate in pH 5.5 medium, but did not grow with N$_2$O as electron acceptor under the same incubation conditions (Supplementary Fig. 6B). These observations corroborate the genomic analysis that the N$_2$O-reducing *Desulfosporosinus* sp. lacks the ability to perform dissimilatory sulfate reduction. Based on phylogenetic and physiologic features, the N$_2$O-reducer in culture EV

represents a novel *Desulfosporosinus* species, for which the name *Desulfosporosinus nitrosoreducens* strain PR is proposed (https://seqco.de/i:32619).

### Genetic underpinning of N$_2$O reduction in *Desulfosporosinus nitrosoreducens* strain PR

The strain PR genome harbors a single *nosZ* gene affiliated with clade II (Fig. 4). Independent branch placement of the strain PR NosZ on the clade II NosZ tree suggests an ancient divergence; a finding supported by NosZ Amino acid Identity (AI) relative to the Average Amino acid Identity (AAI) value of the closest matching NosZ-encoding genome. Specifically, comparisons between the proteins encoded on the genomes of *Desulfosporosinus nitrosoreducens* strain PR and *Desulfosporosinus meridiei* showed genus-level AAI relatedness (i.e., AAI 73.83%), which was significantly higher than the AI of the encoded NosZ (i.e., AI 44%), indicating fast evolution of this protein and/or horizontal *nosZ* acquisition from a distant relative (Figs. 3 and 4). The NosZ of *Desulfosporosinus nitrosoreducens* strain PR is slightly more similar (AI: 45%) to the NosZ of the distant relative *Desulfotomaculum ruminis*.

Comparative analysis of the strain PR *nos* gene cluster with bacterial and archaeal counterparts corroborated characteristic clade II features, including a Sec translocation system, genes encoding cytochromes and an iron-sulfur protein, and a *nosB* gene located

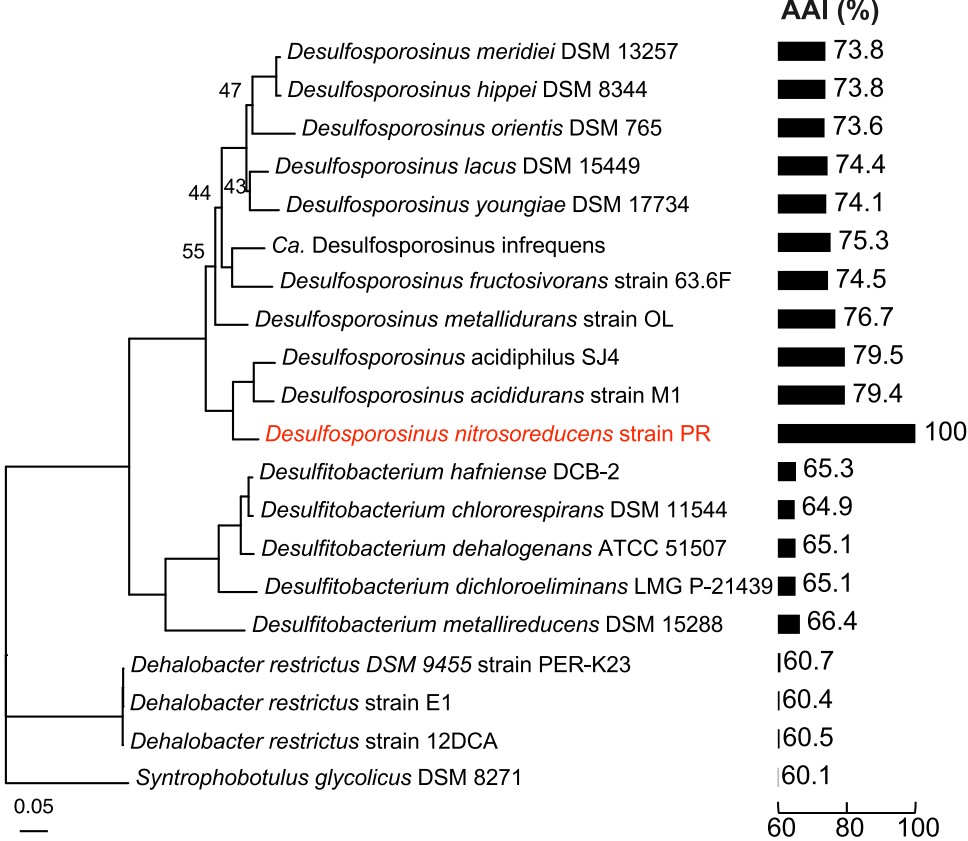

**Fig. 3 | Phylogenomic and Average Amino acid Identity (AAI) analyses indicate that the N$_2$O-reducing strain PR in co-culture EV represents a new species of the genus *Desulfosporosinus*.** Phylogenomic analysis was based on 120 conserved marker genes and included *Peptococcaceae* genomes available from NCBI. Bootstrap values higher than 90 are not displayed. The scale bar indicates 0.05 nucleotide substitution per site. Bar plots display the genome-wide AAI (%) between the N$_2$O-reducing *Desulfosporosinus nitrosoreducens* and related isolates with sequenced genomes. Source data are provided as a Source Data file.

immediately downstream of *nosZ* (Fig. 5). *nosB* encodes a transmembrane protein of unknown function and has been found on clade II, but not clade I *nos* clusters. The *nos* gene clusters of closely related taxa (e.g., *Desulfosporosinus meridiei*, *Desulfitobacterium dichloroeliminans*, *Desulfitobacterium hafniense*) show similar organization; however, differences were observed in the *nos* gene cluster of *Desulfosporosinus nitrosoreducens* strain PR. Specifically, the genes encoding an iron-sulfur cluster protein and cytochromes precede *nosZ* in *Desulfosporosinus meridiei*, but are located downstream of two genes encoding proteins of unknown functions in strain PR (Fig. 5). Of note, among the microbes with *nos* operons and included in the analyses, only *Desulfosporosinus nitrosoreducens* and *Nitratiruptor labii*[43], both with a clade II *nos* cluster, were experimentally validated to grow with N$_2$O below pH 6.

## Genomic insights for a commensalistic relationship
Functional annotation of the *Serratia* sp. and the *Desulfosporosinus nitrosoreducens* strain PR genomes was conducted to investigate the interspecies interactions (Fig. 6). A *btsT* gene encoding a specific, high-affinity pyruvate/proton symporter[44] and genes implicated in pyruvate fermentation (i.e., *pflAB*, *poxB*) are present on the *Serratia* genome, but are missing on the strain PR genome, consistent with the physiological characterization results. *fdhC* genes encoding a formate transporter are present on both genomes, but only the strain PR genome harbors the *fdh* gene cluster encoding a formate dehydrogenase complex (Supplementary Fig. 7), consistent with the observation that the *Serratia* sp. excretes formate, which strain PR utilizes as electron donor for N$_2$O reduction (Supplementary Fig. 1). Gene clusters encoding two different Ni/Fe-type hydrogenases (i.e., *hyp* and *hya* gene clusters)

(Supplementary Fig. 7) and a complete *nos* gene cluster (Fig. 5) are present on the strain PR genome, but not on the *Serratia* sp. genome.

Based on the KEGG[45] and Uni-Prot databases[46], the *Serratia* genome contains the biosynthetic pathways (100% completeness) for aspartate, lysine, threonine, tryptophan, isoleucine, serine, leucine, valine, glutamate, arginine, proline, methionine, tyrosine, cysteine, and histidine. In contrast, only aspartate and glutamate biosynthesis are predicted to be complete on the strain PR genome, whereas the completeness level for biosynthetic pathways of other amino acids was below 80%. The *Serratia* genome encodes a complete set of TCA cycle enzymes, indicating the potential for forming aspartate and glutamate via transamination of oxaloacetate and α-ketoglutarate. In contrast, the strain PR genome lacks genes encoding malate dehydrogenase, citrate synthase, and aconitate hydratase, indicative of an incomplete TCA cycle. Therefore, strain PR is deficient of de novo formation of precursors for glutamate, aspartate, alanine, and related amino acids[47]. A high-affinity amino acid transport system was found on the strain PR genome (Supplementary Fig. 7), suggesting this bacterium can efficiently acquire extracellular amino acids to meet its nutritional requirements.

## Discussion
A few studies reported limited N$_2$O reduction activity in acidic microcosms, but enrichment cultures for detailed experimentation were not obtained[24,48,49]. Possible explanations for the observed N$_2$O consumption in acidic microcosms include residual activity of existing N$_2$O-reducing biomass (i.e., cells synthesized NosZ during growth with N$_2$O as respiratory electron acceptor at a permissible pH show NosZ activity at lower pH; however, no synthesis of new NosZ occurs at

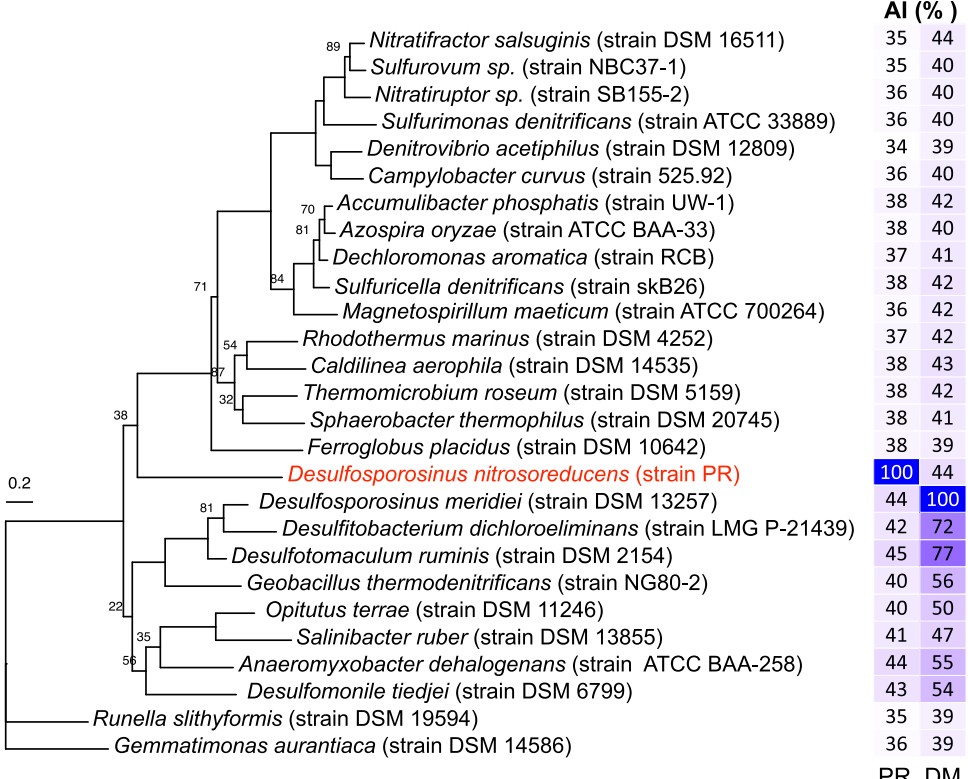

**Fig. 4 | Relatedness and similarity of the clade II NosZ of *Desulfosporosinus nitrosoreducens* strain PR to representative clade II NosZ.** The tree represents a phylogenetic reconstruction of select clade II NosZ protein sequences. The clade II NosZ of *Gemmatimonas aurantiaca* was used to root the tree. The scale bar indicates 0.2 amino acid substitution per site. Numbers at nodes are bootstrap values smaller than 90. The two-column heatmap shows the AAI values between the NosZ of strain PR (PR) and *Desulfosporosinus meridiei* (DM) to other clade II NosZ sequences, with the darker shades of blue indicating higher percent AAI values. Source data are provided as a Source Data file.

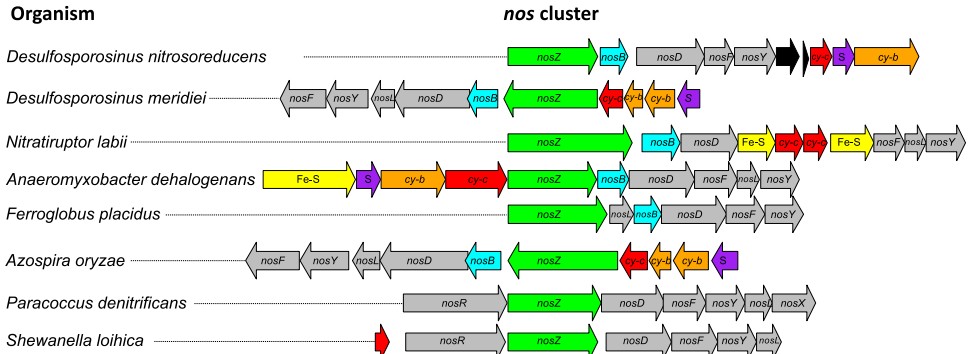

**Fig. 5 | Comparison of representative *nos* clusters.** Included are clade II *nos* clusters encoding select NosZ shown in Fig. 4 and select clade I *nos* clusters of bacteria with confirmed N₂O reduction activity at circumneutral pH (Supplementary Table 4). The colored arrows represent genes with different functions and indicate orientation and approximate length. Green, *nosZ*; gray, *nos* accessory genes (i.e., *nosD*, *nosF*, *nosY*, *nosL*, *nosX* and *nosR*); yellow, genes encoding iron-sulfur (Fe-S) proteins; purple, genes encoding Rieske iron-sulfur proteins (S); orange (cy-b) and red (cy-c), genes encoding b-type and c-type cytochromes, respectively; cyan, *nosB* genes encoding transmembrane proteins characteristic for clade II *nos* operons; black, genes of unknown function. *Desulfosporosinus* and *Desulfitobacterium* spp., *Nitratiruptor* and *Nitratifractor* spp., and *Paracoccus* and *Bradyrhizobium* spp. share similar *nos* cluster architectures, respectively, and representative clusters are shown. Source data are provided as a Source Data file.

acidic pH), or the presence of microsites on soil particles where solid phase properties influence local pH, generating pH conditions not captured by bulk aqueous phase pH measurements[27,50,51]. Soil slurry microcosms providing such microsites with favorable (i.e., higher) pH conditions can give the false impression of low pH N₂O consumption. Removal of solids during transfers eliminates this niche, exposing microorganisms to bulk phase pH, a plausible explanation for the difficulty establishing N₂O-reducing transfer cultures under acidic conditions. Our work with acidic tropical soils highlights another crucial issue, specifically the choice of carbon source for the successful transition from microcosms to soil-free enrichment cultures. Lactate sustained N₂O reduction in pH 4.5 Luquillo tropical soil microcosms, but transfer cultures commenced N₂O reduction only when pyruvate substituted lactate. Lactate has a higher p$K_a$ value than pyruvate (3.8 versus 2.45), indicating that a larger fraction of protonated, and potentially toxic, lactic acid exists at pH 4.5[52]. As discussed above, in soil microcosms, particles with ion exchange capacity (i.e., microsites) can suppress inhibitory effects of protonated organic acids, a possible

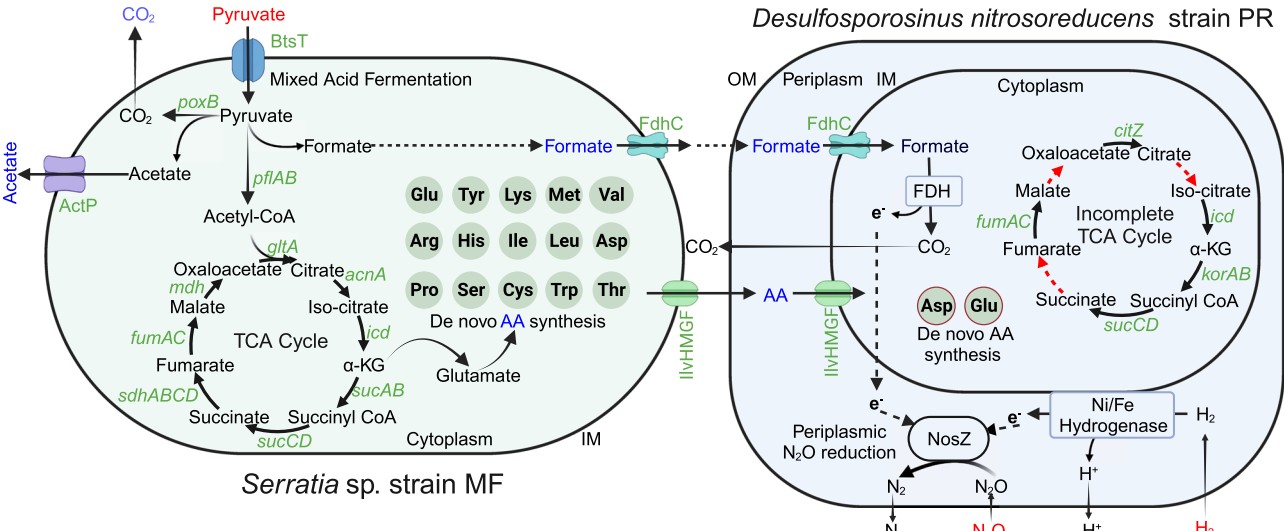

**Fig. 6 | Proposed interspecies cross-feeding interactions in co-culture EV.** Genes involved in mixed acid fermentation are only found on the *Serratia* sp. strain MF genome, and genes involved in periplasmic $N_2O$ reduction are exclusive to *Desulfosporosinus nitrosoreducens* strain PR. External substrates (i.e., pyruvate, $H_2$, $N_2O$) provided to co-culture EV are shown in red font, and metabolites produced by *Serratia* are shown in blue font. Fifteen versus two complete amino acid biosynthesis pathways are present on the *Serratia* sp. strain MF and *Desulfosporosinus nitrosoreducens* strain PR genomes, respectively. Strain PR has an incomplete TCA cycle, and the red dashed arrows indicate the absence of the corresponding genes. TCA cycle tricarboxylic acid cycle, AA amino acids, FDH formate dehydrogenase complex, NosZ nitrous oxide reductase, OM outer membrane, IM inner membrane. Created with BioRender.com.

explanation why lactate supported $N_2O$ reduction in the microcosms but not in the enrichment cultures.

Fifteen repeated transfers with $N_2O$, pyruvate, and $H_2$ yielded a co-culture comprising a *Serratia* sp. and a *Desulfosporosinus* sp. The rapid enrichment of a co-culture was surprising considering that pyruvate and $H_2$ are substrates for many soil microbes. $N_2O$ was the sole electron acceptor provided to the defined basal salt medium, with some $CO_2$ being formed during pyruvate fermentation (Phase I); however, no evidence was obtained for $H_2$-driven $CO_2$ reduction to acetate or to methane. In co-culture EV, the initial dose of $N_2O$ resulted in an aqueous concentration of 2 mM, substantially higher than the reported inhibitory constants for corrinoid-dependent microbial processes[6–8], and both $CO_2/H_2$ reductive acetogenesis and hydrogenotrophic methanogenesis would not be expected to occur in the enrichment cultures, a prediction the analytical measurements support.

Available axenic and mixed denitrifying cultures obtained from circumneutral pH soils reduce $N_2O$ at circumneutral pH, but not under acidic pH conditions[27,38,53]. *Rhodanobacter* sp. strain C01, a facultative anaerobe isolated from acidic (pH 3.7) soil was reported to reduce $N_2O$ at pH 5.7[53]; however, growth with $N_2O$ at pH 5.7 was not demonstrated, and it is possible the observed $N_2O$ reduction activity occurred at higher pH (Supplementary Fig. 8). Characterization of *Nitratiruptor labii*, a facultative anaerobic, strictly chemolithoautotrophic, halophilic deep-sea vent thermophile with a pH optimum of 6.0, provided some evidence for $N_2O$ reduction activity at pH 5.4, but not at pH 5.2[43]. The discovery and cultivation of co-culture EV comprising *Desulfosporosinus nitrosoreducens* strain PR provides unambiguous evidence that a soil bacterium can grow with $N_2O$ as electron acceptor at pH 4.5. Interestingly, strain PR reduces $N_2O$ between pH 4.5 and 6.0, but no $N_2O$ reduction was observed at or above pH 7. This finding implies that *Desulfosporosinus nitrosoreducens* cannot be enriched with $N_2O$ as electron acceptor at or above pH 6.5, suggesting the maintenance of acidic pH conditions during enrichment is crucial for the cultivation of microorganisms capable of low pH $N_2O$ reduction. Apparently, pH selects for distinct groups of $N_2O$ reducers, with prior research focused on facultative anaerobic, denitrifying isolates obtained at circumneutral pH. The discovery of *Desulfosporosinus nitrosoreducens* strain PR lends credibility to the hypothesis that the diverse *nosZ* genes

found in acidic soil metagenomes[36] may indeed be functional. Of note, *nosZ* genes in acidic soils are often found on the genomes of strict anaerobes[37], suggesting that diverse anaerobic bacteria capable of low pH $N_2O$ reduction await discovery. *Desulfosporosinus nitrosoreducens* strain PR sequences were rare in the soil metagenome suggesting that this bacterium was not abundant at the time of sampling, but low abundance members of a community can drive relevant ecosystem processes[54]. Time series sampling would be needed to reveal the in situ population dynamics. The cultivation of strain PR provides a blueprint for unraveling a largely unknown diversity of low pH $N_2O$ reducers and exploring the geochemical parameters that govern this process in acidic soils.

*Desulfosporosinus nitrosoreducens* strain PR possesses a clade II *nos* gene cluster similar to those found in neutrophilic clade II $N_2O$ reducers without clearly distinguishing features based on gene content and synteny (Fig. 5). Experimental work with *Paracoccus denitrificans*, a model organism harboring a clade I *nosZ* and used for studying denitrification to $N_2$, has led to plausible explanations why acidic pH impairs $N_2O$ reduction activity[30]. For example, acidic pH may hinder the binding of $Cu^{2+}$ to the highly conserved histidine residues in the $Cu_A$ and/or $Cu_Z$ sites, implying that NosZ from bacteria capable of low pH $N_2O$ reduction should have altered $Cu_A$ and $Cu_Z$ sites. $Cu_A$ is involved in electron transfer and the $CX_2FCX_3HXEM$ motif was 100% conserved (Supplementary Fig. 9A)[55,56]. The $Cu_Z$ site lacks a conserved motif but has seven characteristic histidine residues with 100% conservation (Supplementary Fig. 9 B). An alignment of curated NosZ sequences, including NosZ of *Desulfosporosinus nitrosoreducens* strain PR, revealed that both clade I and clade II NosZ share 100% conservation of $Cu_A$ and $Cu_Z$ features.

NosZ is a periplasmic enzyme with the mode of secretion differing between clade I versus clade II NosZ organisms. Clade II NosZ follow the general secretion route known as the Sec-pathway, which translocates unfolded proteins across the cytoplasmic membrane. In contrast, clade I NosZ are translocated in their folded state via the Twin-arginine pathway (Tat-pathway)[57]. *nosB*, a gene encoding a transmembrane protein of unknown function, has been exclusively found associated with clade II *nos* clusters (Fig. 5)[40,58]. To what extent *nos* cluster auxiliary gene content and the secretion pathway influence the

pH response of NosZ is unclear and warrants further genetic/biochemical studies. Other factors relevant for $N_2O$ reduction at acidic pH include the organism's ability to cope with the potential toxicity of protonated organic acids and to maintain pH homeostasis[59,60]. The *Desulfosporosinus nitrosoreducens* strain PR genome harbors multiple genes associated with DNA repair and potassium transport, suggesting this bacterium can respond to pH stress. These observations suggest that organismal adaptations to low pH environments play a role, but future research should explore if specific features of NosZ from acidophiles enable $N_2O$ reduction activity under acidic conditions.

Soils harbor diverse microbial communities with intricate interaction networks that govern soil biogeochemical processes[61], including $N_2O$ turnover, and define the functional dynamics of microbiomes[62,63]. Interspecies cooperation between bacteria can enhance $N_2O$ reduction via promoting electron transfer[64], the provision of essential nutrients (as demonstrated in co-culture EV), or limit $N_2O$ reduction due to competition for electron donor(s) or metal cofactors (i.e., copper)[65,66]. Metabolomic workflows revealed that *Serratia* sp. strain MF excretes amino acids during growth with pyruvate, which *Desulfosporosinus nitrosoreducens* strain PR requires to initiate $N_2O$ reduction, a finding supported by genome functional predictions (i.e., 15 complete amino acid biosynthesis pathways in *Serratia* sp. strain MF versus only two complete amino acid biosynthesis pathways in strain PR). Interspecies interactions based on amino acid auxotrophies have been implicated in shaping dynamic anaerobic microbial communities, bolster community resilience, and thus promote functional stability[62]. Other microbes can potentially fulfill the nutritional demands of *Desulfosporosinus nitrosoreducens*, and the observed commensalism between the *Serratia* sp. and *Desulfosporosinus nitrosoreducens* strain PR might have developed coincidentally during the enrichment process.

Members of the genus *Desulfosporosinus* have been characterized as strictly anaerobic sulfate reducers with the capacity to grow autotrophically with $H_2$, $CO_2$, and sulfate, or, in the absence of sulfate, with pyruvate[67]. Most characterized *Desulfosporosinus* spp. show optimum growth at circumneutral pH (~7) conditions, except for the acidophilic isolates *Desulfosporosinus metallidurans*, *Desulfosporosinus acidiphilus*, *Desulfosporosinus acididurans*, and *Desulfosporosinus* sp. strain I2, which perform sulfate reduction at pH 4.0, 3.6, 3.8, and 2.6, respectively[42,68–70]. Among the 10 *Desulfosporosinus* species with sequenced genomes, only the neutrophilic *Desulfosporosinus meridiei* (DSM 13257) carries a *nos* gene cluster[40], but its ability to reduce $N_2O$ has not been demonstrated. *Desulfosporosinus nitrosoreducens* strain PR lacks the hallmark feature of sulfate reduction and is the first acidophilic, strict anaerobic soil bacterium capable of growth with $N_2O$ as electron acceptor at pH 4.5, but not at or above pH 7. Strain PR couples $N_2O$ reduction and growth at pH 4.5 with the oxidation of $H_2$ or formate, and our experimental efforts with co-culture EV could not demonstrate the utilization of other electron donors. The four characterized acidophilic representatives of the genus *Desulfosporosinus* show considerable versatility, and various organic acids, alcohols, and sugars, in addition to $H_2$, support sulfate reduction[42,68,69]. The utilization of $H_2$ as electron donor appears to be a shared feature among *Desulfosporosinus* spp., and two or more gene clusters encoding hydrogenase complexes were found on the available *Desulfosporosinus* genomes[71,72].

Escalating usage of N fertilizers to meet societal demands for agricultural products accelerates N cycling and soil acidification is predicted to increase $N_2O$ emissions. Liming is commonly employed to ameliorate soil acidity, a practice considered beneficial for curbing $N_2O$ emissions based on the assumption that microbial $N_2O$ reduction is favored in circumneutral pH soils[32,38,48,73]. Our findings demonstrate that soil harbors microorganisms (e.g., *Desulfosporosinus nitrosoreducens* strain PR) that utilize $N_2O$ as growth-supporting electron acceptor between pH 4.5 and 6.0. Metagenomic surveys have shown that bacteria capable of low pH $N_2O$ reduction are not limited to acidic

tropical soils, and are more broadly distributed in terrestrial ecosystems[37]. Apparently, acidophilic respiratory $N_2O$ reducers exist in acidic soil and have the potential to mitigate $N_2O$ emissions. Recent efforts have shown success in substantially reducing $N_2O$ emissions from circumneutral and acidic field soils treated with organic waste containing the clade II $N_2O$-reducer *Cloacibacterium* sp. CB-01[74]. The discovery of a naturally occurring acidophilic soil bacterium that couples $N_2O$ consumption to growth between pH 4.5-6.0 offers new opportunities to tackle the $N_2O$ emission challenge and develop knowledge-based management strategies to reduce (i.e., control) $N_2O$ emissions from acidic agricultural soils. Curbing undesirable $N_2O$ emissions at the field scale would allow farmers to further reduce their greenhouse gas emissions footprint and potentially earn carbon credits.

## Methods

### Soil sampling locations and microcosms
Soil samples were collected in August 2018 at the El Verde research station in the El Yunque Natural Forest in Puerto Rico[36]. The measured soil pH was 4.45 and characteristic for the region. Vertical distance of the El Verde research station to mean sea level is 434 meters. Fresh soil materials from 9 to 18 cm depth were used to establish pH 4.5 laboratory microcosms that were amended with $N_2O$ and lactate[37].

### Enrichment process
Transfer cultures were established in 160-mL glass serum bottles containing 100 mL of anoxic, completely synthetic, defined basal salt medium prepared with modifications[75]. The mineral medium consisted of (g L$^{-1}$): NaCl (1.0); $MgCl_2$•6$H_2O$ (0.5); $KH_2PO_4$ (7.0); $NH_4Cl$ (0.3); KCl (0.3); $CaCl_2$•2$H_2O$ (0.015); L-cysteine (0.031) or dithiothreitol (0.15). The medium also contained 1 mL of a trace element solution, 1 mL Se/Wo solution, and 0.25 mL resazurin solution (0.1% w/w). The trace element solution contained (mg L$^{-1}$): $FeCl_2$•4$H_2O$ (1,500); $CoCl_2$•6$H_2O$ (190); $MnCl_2$•4$H_2O$ (100); $ZnCl_2$ (70); $H_3BO_3$ (6); $Na_2MoO_4$•2$H_2O$ (36); $CuCl_2$•2$H_2O$ (2); and 10 mL HCl (25% solution, w/w). The Se/Wo solution consisted of (mg L$^{-1}$): $Na_2SeO_3$•5$H_2O$ (6); $NaWO_4$•2$H_2O$ (8), and NaOH (500). The serum bottles with $N_2$ headspace were sealed with butyl rubber stoppers (Bellco Glass, Vineland, NJ, USA) held in place with aluminum crimp caps. Following autoclaving, the measured medium pH ranged between 4.27 to 4.35. All subsequent amendments to the cultivation vessels used sterile plastic syringes and needles to augment the medium with aqueous, filter-sterilized (0.2 μm polyethersulfone membrane filters, Thermo Fisher Scientific, Waltham, MA, USA) stock solutions and undiluted gases[76]. Ten mL of $N_2O$ gas (416 μmol, 4.16 mM nominal; 99.5%) was added 24 hours prior to inoculation. The bottles were inoculated (1%, v/v) from an El Verde microcosm (established in 160 mL glass serum bottles containing 100 mL of basal salt medium and ~2 g [wet weight] of soil) showing $N_2O$ reduction activity[37]. The microcosm was manually shaken before 1 mL aliquots were transferred with a 3-mL plastic syringe and a 2-gauge needle. Initial attempts to obtain solid-free enrichment cultures with 5 mM lactate as carbon source and electron donor showed no $N_2O$ reduction activity. The following substrates were subsequently tested in the transfer cultures: 5 mM propionate, 20 mM pyruvate, 20 mM pyruvate plus 10 mL (416 μmol, 4.16 mM nominal) hydrogen ($H_2$), 1 mM formate plus 1 mM acetate and 5 mL (208 μmol, 2.08 mM nominal) $CO_2$, and 0.1 or 10 g L$^{-1}$ yeast extract. Subsequent transfers (3%, v/v) used medium supplemented with 0.5 or 2.5 mM pyruvate and 10 mL $H_2$, and occurred when the initial dose of 10 mL $N_2O$ had been consumed. All culture vessels were incubated in upright position at 30 °C in the dark without agitation.

### Microbial community analysis
16S rRNA gene amplicon sequencing was performed on samples collected from 6$^{th}$-generation transfer cultures following complete $N_2O$

consumption, and 9th-generation transfer cultures following complete pyruvate consumption (Phase I) and complete N$_2$O consumption (Phase II). Cells from 1 mL of culture suspension samples were collected by centrifugation (10,000 x g, 20 min, 4 °C), and genomic DNA was isolated from the pellets using the DNeasy PowerSoil Kit (Qiagen, Hilden, Germany). 16S rRNA gene-based amplicon sequencing was conducted at the University of Tennessee Genomics Core following published procedures[77]. Primer set 341F-785R and primer set 515F-805R were used for amplicon sequencing of DNA extracted from 6th and 9th generation transfer cultures, respectively[78].

Analysis of amplicon reads was conducted with nf-core/ampliseq v2.3.1 using Nextflow[79]. Software used in nf-core/ampliseq was containerized with Singularity v3.8.6[80]. Amplicon read quality was evaluated with FastQC v0.11.9[81] and primer removal used Cutadapt v3.4[82]. Quality control including removal of sequences with poor quality, denoising, and chimera removal was performed, and amplicon sequence variants (ASVs) were inferred using DADA2[83]. Barrnap v0.9 was used to discriminate rRNA sequences as potential contamination[84]. ASVs were taxonomically classified based on the Silva v138.1 database[85]. Relative and absolute abundances of ASVs were calculated using Qiime2 v2021.8.0[86]. Short-read fragments of the El Verde soil metagenome representing 16S rRNA genes were identified and extracted using Parallel-Meta Suite v3.7[87].

## Isolation efforts

Following 15 consecutive transfers, 100 µL cell suspension aliquots were serially diluted in basal salt medium and plated on tryptic soy agar (TSA, MilliporeSigma, Rockville, MD, USA) medium. Colonies with uniform morphology were observed, and a single colony was transferred to a new TSA plate. This process was repeated three times before a single colony was transferred to liquid basal salt medium (pH 4.5) amended with 2.5 mM pyruvate, 416 µmol N$_2$O, and 416 µmol H$_2$. Following growth, DNA was extracted for PCR amplification with general bacterial 16S rRNA gene-targeted primer pair 8F-1541R[88] (Integrated DNA Technologies, Inc.,[IDT] Coralville, IA, USA), and Sanger sequencing of both strands yielded a 1471-bp long 16S rRNA gene fragment.

Efforts to isolate the N$_2$O reducer applied the dilution-to-extinction principle[75]. Ten-fold dilution-to-extinction series used 20 mL glass vials containing 9 mL of basal salt medium and 0.8% (w/v) low melting agarose (MP Biomedicals, LLC., Solon, OH) with a gelling temperature below 30 °C[75]. Each glass vial received 2.5 mM pyruvate, 1 mL (41.6 µmol, 4.16 mM nominal) H$_2$ and 1 mL (41.6 µmol, 4.16 mM nominal) N$_2$O following heat sterilization. Parallel $10^{-1}$ to $10^{-10}$ dilution-to-extinction series were established in liquid basal salt medium without low melting agarose, which were used to inoculate the respective soft agar dilution vials. The same dilution-to-extinction procedure was performed in liquid medium and soft agar dilution vials with the 15-amino acid mixture (Supplementary Table 1) substituting pyruvate. Additional attempts to isolate the N$_2$O reducer used solidified (1.5% agar, w/v) basal salt medium. A 1-mL sample of a 15th-generation transfer culture that actively reduced N$_2$O was 10-fold serially diluted in liquid basal salt medium, and 100 µL of cell suspension aliquots were evenly distributed on the agar surface. The plates were incubated under an atmosphere of N$_2$/H$_2$/N$_2$O (8/1/1, v/v/v), and colony formation was monitored every 2 weeks over a 6-month period. Following the isolation of the *Serratia* sp., a two-step approach was tested to isolate the N$_2$O reducer. First, the axenic *Serratia* sp. was grown in defined basal salt medium amended with 2.5 mM pyruvate as the sole substrate. Following complete consumption of pyruvate, the supernatant (i.e., spent medium) was filter-sterilized and transferred to sterile 20 mL glass vials inside an anoxic chamber (N$_2$/H$_2$, 97/3, v/v) (Coy Laboratory Products, Inc., Grass Lake, MI, USA). The vials received 1 mL H$_2$ and 1 mL N$_2$O, and were inoculated from a $10^{-1}$ to $10^{-10}$ serial dilution series of co-culture EV comprising the pyruvate-fermenting

*Serratia* sp. and the N$_2$O-reducing *Desulfosporosinus* sp. This approach tested if the spent medium contains growth factors (i.e., amino acids) that met the nutritional requirement of the N$_2$O-reducing *Desulfosporosinus* sp., without the need for pyruvate addition and associated growth of the *Serratia* sp. Based on the observation that the N$_2$O-reducing *Desulfosporosinus* sp. is a spore former (Supplementary Fig. 10), co-culture EV bottles that had completely consumed pyruvate and N$_2$O were heated to 60 °C or 80 °C for 30 minutes, and cooled to room temperature before serving as inocula (10%, v/v) of fresh medium bottles containing the 15-amino acid mixture, 10 mL H$_2$, and 10 mL N$_2$O.

## Quantitative PCR (qPCR)

A SYBR Green qPCR assay targeting the 16S rRNA gene of the *Serratia* sp., and a TaqMan qPCR assay targeting the 16S rRNA gene of the *Desulfosporosinus* sp. were designed using Geneious Prime (Supplementary Table 2). Probe and primer specificities were examined by in silico analysis using the Primer-BLAST tool[89], and experimentally confirmed using 1538 bp- and 1467 bp-long synthesized linear DNA fragments (IDT) of the respective complete 16S rRNA genes of the *Serratia* sp. and the *Desulfosporosinus* sp., respectively. For enumeration of *Serratia* 16S rRNA genes, 25 µL qPCR tubes received 10 µL 1X Power SYBR Green, 9.88 µL UltraPure nuclease-free water (Invitrogen, Carlsbad, CA, USA), 300 nM of each primer, and 2 µL template DNA. For enumeration of *Desulfosporosinus* 16S rRNA genes, the qPCR tubes received 10 µL TaqMan Universal PCR Master Mix (Life Technologies, Carlsbad, CA, USA), 300 nM of TaqMan probe (5′−6FAM-AAGCTGT-GAAGTGGAGCCAATC-MGB-3′) (Thermo Fisher Scientific), 300 nM of each primer, and 2 µL template DNA[90]. All qPCR assays were performed using an Applied Biosystems ViiA 7 system (Applied Biosystems, Waltham, MA, USA) with the following amplification conditions: 2 min at 50 °C and 10 min at 95 °C, followed by 40 cycles of 15 sec at 95 °C and 1 min at 60 °C. The standard curves were generated using 10-fold serial dilutions of the linear DNA fragments carrying a complete sequence of the *Serratia* sp. (1,538 bp) or the *Desulfosporosinus* sp. (1467 bp) 16S rRNA gene, covering the 70- and 72-bp qPCR target regions, respectively.

The qPCR standard curves established with the linear DNA fragments carrying complete *Serratia* sp. or *Desulfosporosinus* sp. 16S rRNA genes had slopes of −3.82 and −3.404, y-intercepts of 37.408 and 34.181, R$^2$ values of 0.999 and 1, and qPCR amplification efficiencies of 82.7% and 96.7%, respectively. The linear range spanned 1.09 to 1.09 ×$10^8$ gene copies per reaction with a calculated detection limit of 10.9 gene copies per reaction. The genome analysis revealed single copy 16S rRNA genes on both the *Serratia* sp. and the *Desulfosporosinus* sp. genomes, indicating that the enumeration of 16S rRNA gene estimates cell abundances. The 16S rRNA gene sequences of the *Serratia* sp. and the *Desulfosporosinus* sp. are available under NCBI accession numbers OR076433 and OR076434, respectively.

## Nutritional interactions in the co-culture

To explore the nutritional requirements of the *Desulfosporosinus* sp., a time series metabolome analysis of culture supernatant was conducted. Briefly, the axenic *Serratia* sp. culture was grown in basal salt medium amended with 2.5 mM pyruvate, 4.16 mM (nominal) H$_2$, and 4.16 mM (nominal) N$_2$O. Following a 7-day incubation period, during which pyruvate was completely consumed, the bottles received 1% (v/v) co-culture EV inoculum from a 15th transfer culture. Cell suspension aliquots (1.5 mL) were collected and centrifuged, and the resulting cell-free supernatants were transferred to 2 mL plastic tubes and immediately stored at −80 °C for metabolome analysis. Additional samples assessed the metabolome associated with supernatant of axenic *Serratia* sp. cultures that received 1 mM DTT instead of 0.2 mM L-cysteine as reductant. The results of the metabolome analysis guided additional growth experiments with amino acid mixtures replacing

pyruvate. The 100-fold concentrated aqueous 15-amino acid stock solution contained (g L$^{-1}$): alanine (0.5); aspartate (1); proline (1); tyrosine (0.3); histidine (0.3); tryptophan (0.2); arginine (0.5); isoleucine (0.5); methionine (0.4); glycine (0.3); threonine (0.5); valine (0.9); lysine (1); glutamate (1); serine (0.8). The stock solution was filter-sterilized and stored in the dark at room temperature. Growth of co-culture EV in medium amended with the 15-amino acid mixture increased the pH by no more than 0.3 pH units to a maximum observed pH of 4.6.

## Metagenome sequencing

DNA was isolated from the axenic *Serratia* sp. culture grown with 2.5 mM pyruvate, and the N$_2$O-reducing 15$^{th}$ generation co-culture EV grown on H$_2$, N$_2$O, and the amino acid mixture. Metagenome sequencing was performed at the University of Tennessee Genomics Core using the Illumina NovaSeq 6000 platform. Shotgun sequencing generated a total of 494 and 387 Gbp of raw sequences from the axenic *Serratia* sp. culture and co-culture EV. Metagenomic short-reads were processed using the nf-core/mag pipeline v2.1.0[91]. Short-read quality was evaluated with FastQC v0.11.9, followed by quality filtering and Illumina adapter removal using fastp v0.20.1[92]. Short-reads mapped to the PhiX genome (GCA_002596845.1, ASM259684v1) with Bowtie2 v2.4.2 were removed[93]. Assembly of processed short-reads used Megahit2 v1.2.9[94]. Binning of assembled contigs was conducted with MetaBAT2 v2.15[95], and metagenome-assembled genomes that passed CheckM[96] were selected for further analysis. Protein-coding sequences on both genomes were predicted using MetaGeneMark-2[97] and functional annotation used Blastp[98] against the Swiss-Prot database[46], KEGG[45] and the RAST server[99]. Amino acid biosynthesis completeness was evaluated using KofamKOALA[45].

Metagenomic datasets of El Verde soil and a 15$^{th}$ transfer culture were searched against the *Desulfosporosinus nitrosoreducens* strain PR genome using blastn[98]. The best hits were extracted using an in-house script embedded in the Enveomics collection[100]. A graphical representation of short-reads recruited to the *Desulfosporosinus nitrosoreducens* strain PR genome was generated with BlasTab.recplot2.R. The coverage evenness was assessed based on distribution of high nucleotide identity reads across the reference genome sequences. Nonpareil v3.4.1 using the weighted NL2SOL algorithm was used to estimate the average coverage level of the metagenomic datasets[39]. Metagenome data of the original El Verde soil was downloaded from the European Nucleotide Archive (accession number PRJEB74473). Metagenomic datasets of co-culture EV and the genome of the axenic *Serratia* culture were deposited at NCBI under accession numbers SRR24709127 and SRR24709126, respectively (Supplementary Table 3).

## Comparative analysis of *nos* gene clusters

Available genomes of select N$_2$O reducers were downloaded from NCBI (Supplementary Table 4). Functional annotation of the genomes was conducted using the RAST server. Transmembrane topology of the protein encoded by *nosB*, a gene located immediately adjacent to clade II *nosZ* was verified using DeepTMHMM[101]. Accessory genes associated with the *Desulfosporosinus nitrosoreducens nosZ* were identified using cblaster[102] to perform a gene-cluster level BLAST analysis against *Desulfosporosinus*, *Desulfitobacterium*, and *Anaeromyxobacter* genomes. The *nos* gene clusters were visualized using the gggenes package (https://wilkox.org/gggenes/index.html).

## Phylogenomic analysis

Phylogenomic reconstruction was performed with genomes of the *Desulfitobacteriaceae* family available in the NCBI database (Supplementary Table 5). Conserved marker genes of the 20 genomes were identified and aligned with GTDB-TK[103]. Phylogenetic relationships were inferred based on the alignment of 120 concatenated bacterial marker genes using RAxML-NG[104] with 1000 bootstrap replicates. A best fit evolutionary model was selected based on the result of Modeltest-NG[105]. Calculation of Average Amino acid Identity (AAI) and hierarchical clustering of taxa based on AAI values were conducted with EzAAI[106]. Tree annotation and visualization were performed with the ggtree package[107].

## NosZ phylogenetic analysis

NosZ reference sequences were downloaded from pre-compiled models in ROCker[108]. The NosZ sequence of *Desulfosporosinus nitrosoreducens* strain PR was aligned to the NosZ reference sequences using MAFFT[109], and a maximum likelihood tree was created with RAxML-NG based on the best model from Modeltest-NG. The inferred tree and Amino acid Identity (AI) between *Desulfosporosinus nitrosoreducens* strain PR, *Desulfosporosinus meridiei* and the NosZ reference sequences were visualized using the ggtree package.

## Metabolome analysis

Cell-free samples were prepared[110]. Briefly, 1.5 mL of 0.1 M formic acid in 4:4:2 (v:v:v) acetonitrile:water:methanol was added to 100 μL aliquots of supernatant samples. The tubes were shaken at 4 °C for 20 minutes and centrifuged at 16,200 x g for 5 minutes. The supernatant was collected and dried under a steady stream of N$_2$. The dried extracts were suspended in 300 μL water prior to analysis. For water soluble metabolites, the mass analysis was performed in untargeted mode[111]. The chromatographic separations utilized a Synergi 2.6 μm Hydro RP column (100 Å, 100 mm × 2.1 mm; Phenomenex, Torrance, CA) with tributylamine as an ion pairing reagent, an UltiMate 3000 binary pump (Thermo Fisher Scientific), and previously described elution conditions[110]. The mass analysis was carried out using an Exactive Plus Orbitrap MS (Thermo Fisher Scientific) using negative electrospray ionization and full scan mode. Following the analysis, metabolites were identified using exact masses and retention times, and the areas under the curves (AUC) for each chromatographic peak were integrated using the open-source software package Metabolomic Analysis and Visualization Engine[111,112]. Dynamic changes of metabolites over time were assessed by comparative analysis of AUC values.

## Phenotypic characterization of co-culture EV

To test for autotrophic growth of co-culture EV, pyruvate was replaced by 5 mL (2.08 mM nominal) of CO$_2$ (99.5% purity). All experiments used triplicate cultures, and serum bottles without pyruvate, without H$_2$, without N$_2$O, or without inoculum served as controls. Growth experiments were conducted to determine the responses of the *Serratia* sp. and the *Desulfosporosinus* sp. to pH. Desired medium pH values of 4.5, 5, 6, 7 and 8 were achieved by adjusting the mixing ratios of KH$_2$PO$_4$ and K$_2$HPO$_4$. To achieve pH 3.5, the pH 4.5 medium was adjusted with 5 M hydrochloric acid. Replicate incubation vessels received 10 mL (4.16 mM nominal) N$_2$O and 10 mL (4.16 mM nominal) H$_2$, and 2.5 mM pyruvate, following an overnight equilibration period, 1% (v:v) inocula from the axenic *Serratia* sp. culture or the N$_2$O-reducing co-culture EV, both pregrown in pH 4.5 medium. The replicate cultures inoculated with the *Serratia* sp. were incubated for 14 days, after which three vessels received an inoculum of co-culture EV (1%) to initiate N$_2$O consumption. Three *Serratia* sp. cultures not receiving a co-culture EV inoculum served as controls. Consumption rates of pyruvate and N$_2$O were calculated based on data points representing linear ranges of consumption according to

$$V = \frac{N}{T_1 - T_0} \tag{1}$$

where $V$ represent the consumption rate; $N$ represent the initial amounts of pyruvate or N$_2$O. $T_1$ refers to timepoints when pyruvate or

N$_2$O were completely consumed. $T_O$ for pyruvate consumption refers to day zero (i.e., after inoculation with the axenic *Serratia* sp.). $T_0$ for N$_2$O consumption refers to day 14 following inoculation with co-culture EV, which resulted in a linear decrease of N$_2$O.

## Analytical procedures

N$_2$O, CO$_2$, and H$_2$ were analyzed by manually injecting 100 μL head-space samples into an Agilent 3000 A Micro-Gas Chromatograph (Palo Alto, CA, USA) equipped with Plot Q and molecular sieve columns coupled with a thermal conductivity detector[41]. Aqueous concentrations (μM) were calculated from the headspace partial pressures based on reported Henry's law constants[113] for N$_2$O ($2.4 \times 10^{-4}$), H$_2$ ($7.8 \times 10^{-6}$) and CO$_2$ ($3.3 \times 10^{-4}$) mol (m$^3$ Pa)$^{-1}$ according to

$$H^{cp}RT = \frac{C_a}{C_g} \qquad (2)$$

Where $H^{cp}$ is the Henry's law constant[113], $R$ is the universal gas constant, $T$ is the temperature, $C_g$ is the headspace gas-phase concentration, and $C_{aq}$ is the liquid phase (dissolved) concentration. Five-point standard curves for N$_2$O, CO$_2$ and H$_2$ spanned concentration ranges of 8333 to 133,333 ppmv. Pyruvate, acetate and formate were measured with an Agilent 1200 Series high-performance liquid chromatography (HPLC) system (Palo Alto, CA, USA)[41]. pH was measured in 0.4 mL samples of culture supernatant following removal of cells by centrifugation with a calibrated pH electrode.

## Etymology

*Desulfosporosinus nitrosoreducens* (ni.troso.re.du'cens. nitroso, nitrous oxide (N$_2$O), an oxide of nitrogen and intermediate of nitrogen cycling; L. pres. part. reducencs, reducing; from L. v. reduco, reduce, convert to a different condition; N.L. part. adj. nitrosoreducens, reducing N$_2$O).

## Reporting summary

Further information on research design is available in the Nature Portfolio Reporting Summary linked to this article.

## Data availability

The following databases were used in this study: GTDB v2.2.1, NCBI, SRA, KEGG, RAST, Silva v138.1, Swiss-Prot v2023.05, KEGG v2022.07.6, ROCker v1 (https://rast.nmpdr.org). The sequencing data generated in this study have been deposited in the NCBI database under accession number PRJNA951658. The 16S rRNA gene amplicon sequencing data generated from 6$^{th}$ and 9$^{th}$ generation transfer cultures are deposited under SRA accessions SRR24215177 and SRR24083098. The meta-genome raw data generated from co-culture EV and *Serratia* sp. MF have been deposited under SRA accessions SRR24709127 and SRR24709126. The El Verde soil metagenome raw data were generated in a prior study[36] and deposited in the European Nucleotide Archive under accession number PRJEB74473. The 16S rRNA gene sequences of *Desulfosporosinus nitrosoreducens* strain PR and *Serratia* sp. MF have been deposited under GenBank accession numbers OR076434 and OR076433. The draft genomes of *Desulfosporosinus nitrosoreducens* strain PR and *Serratia* sp. MF are available under GenBank accession numbers GCA_030954495.1 and GCA_030954505.1. The metabolomics raw data have been deposited in the MassIVE database under accession number MSV000094351. Source data are provided with this paper.

## Code availability

Code generated for data processing and the production of figures have been deposited in Zenodo (https://zenodo.org/records/10836320).

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

## Acknowledgements

The authors acknowledge funding through the Dimensions of Biodiversity program of the US National Science Foundation (awards 1831599 to F.E.L. and 1831582 to K.T.K.). GH acknowledges support from the China Scholarship Council and a Graduate Student Research Award from the University of Tennessee, Knoxville (UT). DNA sequencing and metabolome analysis were performed in UT's Genomics Core and the Biological and Small Molecule Mass Spectrometry Core, respectively. All bioinformatic analyses were performed on UT's ISAAC Next Generation cluster. We thank Dr. Irene Sánchez-Andrea, Wageningen University & Research, for making available a culture of *Desulfosporosinus acididurans* strain D for comparative growth studies, Dr. Mircea Podar, Oak Ridge National Laboratory, for help with amplicon sequencing, and Dr. Yanchen Sun and Dr. Yongchao Yin for establishing the N$_2$O-reducing tropical soil microcosm.

## Author contributions

G.H. and F.E.L. conceptualized research and designed experiments. F.E.L. and K.T.K. acquired funding. G.H performed experimental work, including cultivation, phenotypic and molecular characterization, and bioinformatic analyses. G. Chen, Y.X, D.R and C.S. provided technical support. K.T.K. and G. Cha provided guidance on bioinformatic analyses. All authors contributed to data analysis and interpretation, and G.H. and F.E.L. wrote the manuscript with inputs from Y.X., G. Chen, K.T.K. and M.R. F.E.L. supervised the study. All authors read and approved the final version of the manuscript.

## Competing interests

The authors declare no competing interests.
