## [Peer Review File · Nature Communications]

Sustained bacterial N₂O reduction at acidic pHReviewer #1 (Remarks to the Author):

This manuscript describes a stable co-culture of *Serratia* sp. and "Desulfosporosinus nitroso-reducens" from an acidic soil that continued to reduce N₂O at pH 4.5. The authors verified that the *Desulfosporosinus* cells efficiently grew under N₂O as a sole electron acceptor in the co-culture under the acidic condition. *Serratia* sp. cells may provide 15 different amino acids and formate to the *Desulfosporosinus* cells, which was fully supported by genomic and metabolite analyses. The results microbiologically supported the N₂O reduction potential of acidic soils.

Major comments:

The experimental designs, presentation and explanations are well organized throughout this manuscript. In particular, the authors well designed and performed the enriched culture experiments and the corresponding omics analyses. The authors unfortunately could not isolate N₂O-reducing "Desulfosporosinus nitroso-reducens", but cleverly continued their experiments in the co-culture of *Serratia* sp. and "Desulfosporosinus nitroso-reducens". In this regard, I have several questions and comments.

1) The authors proposed a new species "Desulfosporosinus nitroso-reducens" and designated "strain PR" in the manuscript. I wonder this proposal without pure culture of "strain PR".

2) Have the authors isolate "*Serratia* sp. strain MF" in this work?

3) I think that the growth yield data of *Desulfosporosinus* sp. under N₂O as a sole electron acceptor is physiological crucial data in this work as compared with previous research background of acidic N₂O reducers and clade II N₂O reducers. However, the broad readers do not always understand the significance of results. I recommend the new table to emphasize this point probably from supplementary materials.

4) Please let us and readers the efforts to isolate and purify "Desulfosporosinus nitroso-reducens strain PR". The isolation is challenging but might be feasible based on the genome of "Desulfosporosinus nitroso-reducens" and physiological data in this manuscript.

5) Even in the liquid of co-culture, different microbes may physically associate each other in particular under strict anaerobic environments. Have the authors microscopically observe the cell configuration in the co-culture? Cell aggregates or planktonically dispersed cells in the co-culture? How are the cell shapes of *Serratia* sp. and *Desulfosporosinus* sp.?

Minor comments:

L644 (Fig. 1 legend): Please spell out "EV" in Fig. 1 legend. Is this EI Verde Field Soil? Please check the abbreviations of EI Verde Field Soil throughout Fig. 1 legend and text.

Reviewer #2 (Remarks to the Author):

This work is exceptional, pushing the boundaries of our knowledge regarding the reduction of N₂O in acidic soils. The study is exploratory, forming hypotheses and testing them effectively as they arise. The methods are adequate, and the interpretations are flawless.

The reduction of N₂O in acidic soil may seem like a nerdy topic, but it is actually a significant challenge to understand, and it has far-reaching implications for the climate: anthropogenic acidification of soils is a global-wide problem, and one of the consequences is enhanced emissions of N₂O, a strong climate gas, because the bacterial N₂O-reduction in soil is hampered by low pH. This phenomenon is well understood based on the investigations of "canonical" denitrifying bacteria: these organisms are unable to synthesize functional N₂O-reductase (*nosZ*) at pH < 6. Nevertheless, significant N₂O-reduction place in very acid soils, suggesting that there must exist organisms that can synthesize functional *nosZ* at such low pH. Finding such organisms, and

understanding "how they do it" is a prominent task, for two obvious reasons: one is to understand the phenomenon, the second is that this would open an avenue for mitigating the high N₂O emission from acidic soil by bioengineering the microbiota of acid soils.

The authors set out to search for bacteria that can reduce N₂O at low pH, and used enrichment culturing of soil bacteria to find them, starting with an inoculum from an acid tropical forest soil. Through 15 sequential low pH batch enrichments, provided with pyruvate as a C source and N₂O as the sole terminal electron acceptor, they obtained complete dominance by two organisms: a *Serratia* sp which fermented pyruvate, producing formate and H₂, and a *Desulfosporosinus* sp which respired formate/H₂ with N₂O as terminal electron acceptor. They characterized the co-culture physiologically, and were able to demonstrate that *Serratia* provided the N₂O-respiring *Desulfosporosinus* sp both with formate/H₂ and several amino acids which it was unable to synthesize itself. The genomes of the two organisms were assembled based on shotgun sequencing of the co-culture, which revealed that the genome of *Desulfosporosinus* sp. is deficient of de novo formation of precursors for several amino acids.

The authors demonstrate a strong grasp of the issue and correctly assert that previous claims regarding "acidophilic" N₂O-reducing bacterial strains lack rigor.

Unfortunately, all attempts to isolate *Desulfosporosinus* sp failed, probably reflecting that *Serratia* provides it with more than amino acids and formate/H₂.

In conclusion: this deserves being published in NatCom!

That said, I have a few comments to the text:

The discussion is perhaps a bit lengthy and repetitive towards the end (repeating things already present in the introduction). And I miss one issue that should be mentioned in the discussion: The acid soil did reduce N₂O, but could this possibly be ascribed to *Desulfosporosinus* sp, given its extremely low abundance in the soil?

Specific comments on the main text:

Line 48: drop this sentence. It doesn't suit the manuscript to use the vacuous term "soil health." Nor does it suit the manuscript to refer to Lal.

Line 60: I'm not sure that all the papers referred to (29-31) demonstrated N₂O-reduction in soil/slurries with pH < 5.

Line 73: I wonder if this could be rephrased. The paradigm is that the synthesis of functional nosZ is severely hampered by low pH, which explains why acidification enhances N₂O emission. This paradigm fails to explain the fact that N₂O-reduction takes place in very acidic soils, albeit at low rates.

Line 92: perhaps use a few more words to explain "transfer cultures"?

Line 98: Fig 1D

Line 105: where is this shown? Should possibly be added to Supplementary Information

Line 269: Why do you assume that N₂O should be uncoupled to growth at low pH? I see no reason for this.

Line 340-342: I don't agree.

Line 385: Low pH N₂O-reduction

Line 389: Gao et al 2016: the active N₂O-reducing bacteria were located in pelleted organic material with high pH => no evidence of acidophilic N₂O-reduction

Line 643-663 (Fig 1 Legend): define "co-culture EV" clearly in this legend, or where it occurs the first time in the text.

Line 665-685 (Fig 2 Legend): Check the references to the panels and their color codes. The legend seems to refer to an old version of the figure. For example there is no blue and red background in panel A..

Supplementary Information:

Line 28: write the composition

Line 36: how much soil?

Line 112: explain that EV inoculum is inoculum from the 15th enrichment

REVIEWER COMMENTS

Reviewer #1 (Remarks to the Author):

This manuscript describes a stable co-culture of *Serratia* sp. and “*Desulfosporosinus nitroso-reducens*” from an acidic soil that continued to reduce N₂O at pH 4.5. The authors verified that the *Desulfosporosinus* cells efficiently grew under N₂O as a sole electron acceptor in the co-culture under the acidic condition. *Serratia* sp. cells may provide 15 different amino acids and formate to the *Desulfosporosinus* cells, which was fully supported by genomic and metabolite analyses. The results microbiologically supported the N₂O reduction potential of acidic soils.

Major comments:

The experimental designs, presentation and explanations are well organized throughout this manuscript. In particular, the authors well designed and performed the enriched culture experiments and the corresponding omics analyses. The authors unfortunately could not isolate N₂O-reducing “*Desulfosporosinus nitroso-reducens*”, but cleverly continued their experiments in the co-culture of *Serratia* sp. and “*Desulfosporosinus nitroso-reducens*”. In this regard, I have several questions and comments.

1) The authors proposed a new species “*Desulfosporosinus nitroso-reducens*” and designated “strain PR” in the manuscript. I wonder this proposal without pure culture of “strain PR”.

Response. It has become commonplace to assign strain designations to microbes that have been identified via genomic or cultivation approaches but without axenic cultures available. We considered omitting the strain designation, but this would require us to spell out the name *Desulfosporosinus nitroso-reducens*, what would make several sentences cumbersome. We discussed this issue in depth and considered introducing the term “bacterium PR” but decided to keep the strain designation. This seems justified because we have the genome and the co-culture consisting of *Serratia* sp. strain MF and *Desulfosporosinus nitroso-reducens* strain PR.

2) Have the authors isolate “*Serratia* sp. strain MF” in this work?

Response. Yes, the *Serratia* species was isolated and is available in axenic culture. We added a clarifying statement in the revised manuscript (line 136) and included a detailed isolation procedure in Supplementary Information (page 3).

3) I think that the growth yield data of *Desulfosporosinus* sp. under N₂O as a sole electron acceptor is physiological crucial data in this work as compared with previous research background of acidic N₂O reducers and clade II N₂O reducers. However, the broad readers do not always understand the significance of results. I recommend the new table to emphasize this point probably from supplementary materials.

Response. In response to the reviewer’s suggestion, we highlight in the revised text that *Desulfosporosinus nitroso-reducens* reduces N₂O in a growth-linked process at pH 4.5. We also present yield data for bacteria determined under comparable growth conditions. Specifically, we now state that “The growth yield of *Desulfosporosinus nitroso-reducens* with N₂O is on par with growth yields reported for neutrophilic N₂O reducers with clade II *nosZ* such as *Anaeromyxobacter* spp. and *Dechloromonas aromatica* (Sanford et al. 2012, PNAS, <https://doi.org/10.1073/pnas.1211238109>; Yoon et al. 2016, Appl. Environ. Microbiol. <https://doi.org/10.1128/AEM.00409-16>) (line 129). Since yield data for strictly respiratory growth with N₂O as electron acceptor are rare, we do not see a need for a new Supplementary Material table.

4) Please let us and readers the efforts to isolate and purify “Desulfosporosinus nitroso-reducens strain PR”. The isolation is challenging but might be feasible based on the genome of “Desulfosporosinus nitroso-reducens” and physiological data in this manuscript.

Response. We have decade-long expertise with the cultivation of fastidious microorganisms and tried very hard to obtain an axenic culture of *Desulfosporosinus nitroso-reducens* capable of low pH N₂O reduction. Unfortunately, none of our approaches yielded an axenic culture. We employed the dilution-to-extinction principle both in liquid and soft agar medium supplemented with pyruvate or the 15 amino acid mixture plus H₂ and N₂O. We recovered N₂O reduction activity in the 10⁻⁶ liquid dilution vials; however, the *Serratia* was always present. In soft agar dilution vials, NosZ-catalyzed N₂O reduction generated N₂, and the formation of N₂ gas bubbles prevented us from picking isolated colonies (illustrated in Fig. S5). Additionally, we tried to grow the co-culture on solid (1.5% agar) amino acid-containing mineral salt medium incubated under a headspace of N₂/H₂/N₂O (8/1/1, v/v/v) for up to 2 months. We observed uniform colonies, which all represented the *Serratia* sp. as verified by microscopic observation and 16S rRNA gene Sanger sequencing. We also attempted to recover N₂O reduction activity from pasteurized co-cultures based on the observation that *Desulfosporosinus nitroso-reducens* is a spore former; however, we were unable to recover active cultures following pasteurization. We agree that these negative results constitute relevant information, especially for researchers interested in the cultivation and characterization of new isolates. We have expanded the section detailing the unsuccessful isolation efforts. Since not all readers will be interested in such level of detail describing unsuccessful isolation work, we present these efforts in the revised Supplementary Information document (page 3).

5) Even in the liquid of co-culture, different microbes may physically associate each other in particular under strict anaerobic environments. Have the authors microscopically observe the cell configuration in the co-culture? Cell aggregates or planctonically dispersed cells in the co-culture? How are the cell shapes of *Serratia* sp. and *Desulfosporosinus* sp.?

Response. We have regularly performed light microscopic analysis to observe the enrichment process and the co-culture. The analysis of the co-culture revealed two distinct cell morphologies consistent with reported cell shapes of *Serratia* and *Desulfosporosinus*. Evidence for the physical association between *Serratia* and *Desulfosporosinus* cells was not obtained, and the formation of *Serratia-Desulfosporosinus* cell aggregates was not observed. In addition, we grew the axenic *Serratia* sp. culture in defined mineral salt medium amended with pyruvate as the sole substrate, and filter-sterilized the medium following complete consumption of pyruvate. The sterile, spent medium was then supplemented with H₂ and N₂O, and inoculated with the co-culture comprising the *Serratia* sp. and *Desulfosporosinus nitroso-reducens*. The rationale was that the spent medium contains amino acids to meet the nutritional requirements of *Desulfosporosinus nitroso-reducens*, but no pyruvate was present to support the growth of *Serratia*. Indeed, we observed N₂O and H₂ consumption following inoculation indicating activity of *Desulfosporosinus nitroso-reducens*; however, slight growth of *Serratia* also occurred, presumably on the amino acids in the medium or dead *Desulfosporosinus* cells. Although initially promising, this strategy did not yield an axenic culture of the N₂O reducer. The *Serratia* cells are rod shaped, 2-3 μm long as shown in the microscopic image (Fig. 1 A). The *Desulfosporosinus* cells are longer (5-6 μm long) curved rods highlighted by the red arrows in Fig. 1 B. In older cultures, spore formation in *Desulfosporosinus nitroso-reducens* cells was observed (Fig. 1 C).

Fig. 1. Light microscopic images showing the morphologies of *Serratia* sp. strain MF and *Desulfosporosinus nitroso-reducens* strain PR cells. (A) Axenic culture of *Serratia* sp. strain MF. (B) Co-culture EV comprising *Serratia* sp. strain MF and *Desulfosporosinus nitroso-reducens* strain PR. (C) Four-week-old co-culture EV showing spore formation in *Desulfosporosinus nitroso-reducens* strain PR cells.

Minor comments:

L644 (Fig. 1 legend): Please spell out “EV” in Fig. 1 legend. Is this El Verde Field Soil? Please check the abbreviations of El Verde Field Soil throughout Fig. 1 legend and text.

Response. We have carefully reviewed the script for consistent use of abbreviations. Co-culture EV (referring to El Verde, the name of the tropical forest where the original soil sample was collected) was obtained from microcosms established with El Verde soil. We have revised the figure legend to ensure the reader can quickly grasp content without searching for abbreviations in the text.

Reviewer #2 (Remarks to the Author):

This work is exceptional, pushing the boundaries of our knowledge regarding the reduction of N₂O in acidic soils. The study is exploratory, forming hypotheses and testing them effectively as they arise. The methods are adequate, and the interpretations are flawless.

Response. We very much appreciate these positive comments. The experimental work was demanding, and we had to overcome several challenges, and it is gratifying to read such encouraging feedback.

The reduction of N₂O in acidic soil may seem like a nerdy topic, but it is actually a significant challenge to understand, and it has far-reaching implications for the climate: anthropogenic acidification of soils is a global-wide problem, and one of the consequences is enhanced emissions of N₂O, a strong climate gas, because the bacterial N₂O-reduction in soil is hampered by low pH. This phenomenon is well understood based on the investigations of “canonical” denitrifying bacteria: these organisms are unable to synthesize functional N₂O-reductase (*nosZ*) at pH<6. Nevertheless, significant N₂O-reduction place in very acid soils, suggesting that there must exist organisms that can synthesize functional *nosZ* at such low pH. Finding such organisms, and understanding “how they do it” is a prominent task, for two obvious reasons: one is to understand the phenomenon, the second is that this would open an avenue for mitigating the high N₂O emission from acidic soil by bioengineering the microbiota of acid soils.

The authors set out to search for bacteria that can reduce N₂O at low pH, and used enrichment culturing of soil bacteria to find them, starting with an inoculum from an acid tropical forest soil. Through 15 sequential low pH batch enrichments, provided with pyruvate as a C source and N₂O as the sole terminal electron acceptor, they obtained complete dominance by two organisms: a *Serratia* sp

which fermented pyruvate, producing formate and H₂, and a *Desulfosporosinus* sp which respired formate/H₂ with N₂O as terminal electron acceptor. They characterized the co-culture physiologically, and were able to demonstrate that *Serratia* provided the N₂O-respiring *Desulfosporosinus* sp both with formate/H₂ and several aminoacids which it was unable to synthesize itself. The genomes of the two organisms were assembled based on shotgun sequencing of the co-culture, which revealed that the genome of *Desulfosporosinus* sp. is deficient of de novo formation of precursors for several amino acids.

The authors demonstrate a strong grasp of the issue and correctly assert that previous claims regarding "acidophilic" N₂O-reducing bacterial strains lack rigor.

Unfortunately, all attempts to isolate *Desulfosporosinus* sp failed, probably reflecting that *Serratia* provides it with more than amino acids and formate/H₂.

In conclusion: this deserves being published in NatCom!

That said, I have a few comments to the text:

The discussion is perhaps a bit lengthy and repetitive towards the end (repeating things already present in the introduction). And I miss one issue that should be mentioned in the discussion: The acid soil did reduce N₂O, but could this possibly be ascribed to *Desulfosporosinus* sp, given its extremely low abundance in the soil?

Response. We carefully reviewed the script and revised the text avoid redundancies. Specifically, we have condensed the Discussion section by removing redundant information.

The reviewer's 2nd question is a very good one and requires a more elaborate answer. The reviewer is correct that *Desulfosporosinus nitroso-reducens* was not an abundant member of the community based on the soil metagenome data but emerged as the dominant N₂O-reducing bacterium upon laboratory enrichment with N₂O as electron acceptor at pH 4.5. The question emerges if an organism present in low abundance can be responsible for an environmentally relevant process, here the reduction of N₂O at low pH. First, relative abundance data may or may not correlate with a specific process of interest the community is performing. We have done a lot of work with strictly organohalide-respiring bacteria (e.g., *Dehalococcoides*), which are common but minor members of the community that often go undetected in metagenome sequence datasets; however, they carry out the important process of reductive dechlorination. The point is that low abundance populations can carry out relevant ecosystem processes.

Desulfosporosinus nitroso-reducens was enriched under the cultivation conditions applied in our laboratory. A previous study suggested that the original soil harbored multiple taxa capable of low pH N₂O reduction (Sun et al. 2024. <https://www.biorxiv.org/content/10.1101/2023.11.29.569236v1>), suggesting that more than a single species contributes to low pH N₂O reduction in this soil.

Another relevant consideration is the dynamic nature of soil microbiomes, and relative abundances may fluctuate over time. Our soil sampling effort represents a snapshot in time, not allowing us to assess time-resolved population dynamics. Our work generates that basis for conducting time-resolved measurements aimed at deciphering population dynamics and correlating abundance and expression data with *in situ* N₂O fluxes.

In response to the reviewer's question, we have slightly expanded the discussion on this topic (lines 302-305).

Specific comments on the main text:

Line 48: drop this sentence. It doesn't suit the manuscript to use the vacuous term "soil health." Nor does it suit the manuscript to refer to Lal.

Response. We agree that this sentence is not needed.

Line 60: I'm not sure that all the papers referred to (29-31) demonstrated N₂O-reduction in soil/slurries with pH<5.

Response. Yes, thanks for pointing this out. We carefully checked the information provided in the cited papers (i.e., references 29-31). The reviewer is correct that references 30 and 31 were not well chosen. Reference 30 expands the work on a set of microcosms published in another journal, where low pH N₂O reduction was reported. We now include the correct citation. Reference 31 does not demonstrate N₂O reduction activity and has been replaced with a paper from the same research group, which we intended to cite (reference 18 in the revised paper). This reference reports N₂O reduction activity in pH 3.7 to 5.4 soil microcosms.

Line 73: I wonder if this could be rephrased. The paradigm is that the synthesis of functional nosZ is severely hampered by low pH, which explains why acidification enhances N₂O emission. This paradigm fails to explain the fact that N₂O-reduction takes place in very acidic soils, albeit at low rates.

Response. We agree that our statement was not as precise as it should have been, and we rephrased this sentence taking the reviewer's suggestions into consideration (lines 65-66).

Line 92: perhaps use a few more words to explain "transfer cultures"?

Response. Although "transfer cultures" is common terminology for researchers engaged in enrichment and cultivation, we agree that in a manuscript with appeal to a broader audience, such terms should be briefly explained. We rephrased this sentence for clarification (line 85).

Line 98: Fig 1D

Response. Thank you for pointing out this error. We should have referred the reader to Figure 1, not a specific panel of Figure 1.

Line 105: where is this shown? Should possibly be added to Supplementary Information

Response. The data were included in the Supplementary Information, and we now specifically refer the reader to Supplementary Fig. S2.

Line 269: Why do you assume that N₂O should be uncoupled to growth at low pH ? I see no reason for this.

Response. We provide explanations why prior studies have failed to enrich low pH N₂O reducers. Several studies established N₂O-reducing microcosms and/or enrichment cultures at circumneutral pH, which were subsequently exposed to low pH conditions. Initial N₂O reduction was observed; however, the activity was lost upon transfer. A plausible reason is residual N₂O reduction activity from biomass grown at circumneutral pH, but no new growth occurred under acidic pH conditions (i.e., N₂O reduction activity was uncoupled growth). To avoid confusion, we have carefully revised this section for clarity (lines 257-259).

Line 340-342: I don't agree.

Response. Mechanistic understanding of low pH N₂O reduction is lacking. Based on the data we have generated, we suggest that the organisms' ability to adapt to a low pH environment is a key feature of low pH N₂O reducers. The reviewer argues that the features of an N₂O reductase (i.e., NosZ) that works

at acidic pH differ from those of an N₂O reductase that functions at circumneutral pH. At this point, we do not know the specific features that distinguish NosZ in organisms active at low pH versus circumneutral pH. We agree with the reviewer that further mechanistic studies are required. For example, comparative structural analyses of NosZ enzymes from acidophilic and circumneutral N₂O reducers could address this question. In response, we modified this sentence without firmly implying that organismal adaptations allow some bacteria to reduce N₂O at acidic pH (lines 331-333).

Line 385: Low pH N₂O-reduction

Response. Thank you, this oversight was corrected as suggested.

Line 389: Gao et al 2016: the active N₂O-reducing bacteria were located in pelleted organic material with high pH => no evidence of acidophilic N₂O-reduction

Response. The reviewer is correct that Gao et al. 2016 (reference 90) does not demonstrate N₂O reduction at low pH. Our intention was to demonstrate that bioaugmentation to enhance N₂O reduction is a feasible approach. Gao et al. 2016 (reference 90) performed greenhouse pot experiments, but a much more convincing study demonstrating that bioaugmentation can stimulate N₂O reduction at field scale has recently become available (Hiis et al. 2024). We updated the citations and reworded the text to clarify that bioaugmentation to enhance N₂O reduction activity is a feasible approach (lines 376-378).

Line 643-663 (Fig 1 Legend): define “co-culture EV” clearly in this legend, or where it occurs the first time in the text.

Response. We did define co-cultures EV in the text (line 113). To improve comprehension, we added a brief definition to the legend of Figure 1 (line 637).

Line 665-685 (Fig 2 Legend): Check the references to the panels and their color codes. The legend seems to refer to an old version of the figure. For example there is no blue and red background in panel A..

Response. We apologize and thank the reviewer for catching this oversight. Apparently, the background colors disappeared during the conversion from Word to the PDF file format. This issue will be corrected when we upload separate PDF figure files (i.e., the figures are not embedded in a Word document).

Supplementary Information:

Line 28: write the composition

Response. We believe the reviewer suggests that we include the medium recipe in this paragraph. We agree, and we added the medium composition to this paragraph.

Line 36: how much soil?

Response. We agree that this information should have been provided and we now indicate that about 2 g (wet weight) of soil was used for microcosm setup.

Line 112: explain that EV inoculum is inoculum from the 15th enrichment

Response. We agree and we added a brief statement clarifying that the co-culture EV inoculum was derived from a 15th generation transfer culture.